# The Impact of Possible Decadal-Scale Cold Waves on Viticulture over Europe in a Context of Global Warming

**Giovanni Sgubin [1],\*, Didier Swingedouw [1] , Iñaki García de Cortázar-Atauri [2],**
**Nathalie Ollat [3] and Cornelis van Leeuwen [3]**

[1] Environnements et Paléoenvironnements Océaniques et Continentaux (EPOC), University Bordeaux, 33615 Pessac, France
[2] Institut National de la Recherche Agronomique (INRA), US1116-Agroclim, F-84914 Avignon, France
[3] Ecophysiologie et Génomique Fonctionnelle de la Vigne (EGFV), Bordeaux Sciences Agro, University Bordeaux, INRA, ISVV, 33882 Villenave d'Ornon, France
\* Correspondence: giovanni.sgubin@u-bordeaux.fr; Tel.: +33-540-006-193

**Abstract:** A comprehensive analysis of all the possible impacts of future climate change is crucial for strategic plans of adaptation for viticulture. Assessments of future climate are generally based on the ensemble mean of state-of-the-art climate model projections, which prefigures a gradual warming over Europe for the 21st century. However, a few models project single or multiple O(10) year temperature drops over the North Atlantic due to a collapsing subpolar gyre (SPG) oceanic convection. The occurrence of these decadal-scale "cold waves" may have strong repercussions over the continent, yet their actual impact is ruled out in a multi-model ensemble mean analysis. Here, we investigate these potential implications for viticulture over Europe by coupling dynamical downscaled EUR-CORDEX temperature projections for the representative concentration pathways (RCP)4.5 scenario from seven different climate models—including CSIRO-Mk3-6-0 exhibiting a SPG convection collapse—with three different phenological models simulating the main developmental stages of the grapevine. The 21st century temperature increase projected by all the models leads to an anticipation of all the developmental stages of the grapevine, shifting the optimal region for a given grapevine variety northward, and making climatic conditions suitable for high-quality wine production in some European regions that are currently not. However, in the CSIRO-Mk3-6-0 model, this long-term warming trend is suddenly interrupted by decadal-scale cold waves, abruptly pushing the suitability pattern back to conditions that are very similar to the present. These findings are crucial for winemakers in the evaluation of proper strategies to face climate change, and, overall, provide additional information for long-term plans of adaptation, which, so far, are mainly oriented towards the possibility of continuous warming conditions.

**Keywords:** climate change; *Vitis vinifera* L.; general circulation model; EURO-CORDEX; phenological model

## 1. Introduction

The production of high-quality wine represents a valuable cultural and economic patrimony for many local communities all over Europe, notably in France, Italy, and Spain, which together account for about half of the world production [1]. The reputation of currently recognized winegrowing regions mainly results from a complex combination of favorable climatic conditions [2]. Along with particular local soil compositions, typical grape varieties, and the expertise in vineyard management matured and handed down over centuries, specific climatic conditions define the concept of *terroir* [3,4]. Premium

wine production is, in this context, acknowledged by specific certifications in Europe and preserved by regional regulations, like, inter alia, the French AOC (*Appellation d'Origine Controlee*), the Italian DOCG (*Denominazione di Origine Controllata e Garantita*), and the Spanish DO (*Denominación de Origen*). The maintenance and the expansion of the European wine-making heritage is, however, a delicate matter in the context of global warming [5], as the equilibrium between the different climatic conditions may be altered in the future and therefore the *terroir* characteristics.

Temperature is the predominant driver of grapevine (*Vitis vinifera*) growing [6], as it primarily regulates the main phenological phases of the plant, i.e., bud break, flowering, veraison, and maturity, thus characterizing yield and quality parameters. Due to the ongoing climate change, earlier phenological events have been registered in the last decades over most of the traditional vineyards of Europe, e.g., in Bordeaux and Rhone Valley [7,8], northeast Spain [9], northeast Italy [10,11], and Piedmont [12].

By modulating the length of each phenological phase, temperature also plays a central role in determining the fruit composition [13] by outlining the ratio between sugar content and acidity [14], whose equilibrium is essential for high-quality wines [15]. Temperatures that are too high would produce precocious development of the fruit, resulting in wines with high alcohol and low organic acid contents [16]. On the contrary, conditions that are too cool would prevent the complete maturity of the fruits, yielding berries with high acidity, low sugar, and unripe flavors [7,17]. This is one of the reasons why climatic conditions primarily determine the potential for premium wine production in a given region [18,19]. Indeed, in order to accomplish a balanced development of the fruit, maturity in the northern hemisphere should occur between approximatively 10 September and the 10 October [2], thus implying the mean temperature during the growing season needs to be bounded within a narrow range. In [18], temperature limits between approximatively 12 °C and 22 °C were proposed to define suitable growing areas for *Vitis vinifera*, while for individual grapevine varieties, this range is much narrower down to 2 °C, e.g., for Pinot noir.

These temperature thresholds define the suitable climatic conditions for high-quality wine potential, thus identifying specific grapevine varieties for each particular winegrowing region and characterizing the geography of premium wine production. Early ripening grapevines varieties like Chardonnay, Pinot noir, and Riesling are typically cultivated in the northernmost vineyards of Europe, e.g., Germany and U.K., as well as in continental regions, e.g., Champagne, Alsace, and Burgundy (France), and mountains regions, e.g., Trentino Alto Adige (Italy). For their characteristics, these varieties are those classically selected for new plantations in the so-called "cool-climate wine" areas. Average ripening varieties are currently cultivated in the Atlantic sector of Europe, e.g., Merlot in Bordeaux (France), Tempranillo in Rioja (Spain), and Touriga in Douro (Portugal), and in hilly areas of the Mediterranean sector, e.g., Syrah in Rhone Valley (France), Sangiovese in Tuscany (Italy). These varieties potentially risk over-ripening under too-warm climate conditions. Late ripening varieties are currently cultivated in the Mediterranean region, e.g., Grenache in Languedoc (France), Sardinia (Italy), Arangon, and Navarra (Spain), and in the warmest regions of the Atlantic sector, e.g., Cabernet Sauvignon in Bordeaux (France). These varieties are expected to expand to northern regions under climate change.

The general warmer conditions registered all over the Europe in the last decades have already promoted new viticulture areas to emerge beyond 50° N. For example, the vineyard coverage of England and Wales has more than doubled since 2004 according to recent estimations [20]. A similar trend is observed in Denmark [21]. Moreover, warmer conditions, so far, appear to be generally beneficial for many traditional vineyards, since the optimum climate for their typical varieties has been approached. For example, Merlot and Cabernet Sauvignon in Bordeaux have tended to produce larger berry weights and higher sugar to total acid ratios, which corresponded to an increase of vintage rating [22]. This was likely due to earlier veraison dates, which enabled wine-makers to have a larger margin of time to establish when the optimal fruit composition was reached, with the possibility to pick fruits at greater levels of ripeness [7]. However, further warming over these traditional regions

may push climatic conditions beyond the optimum for their typical varieties, which will likely force wineries to adapt and eventually switch to more appropriate varieties for warmer climates [23,24].

For the future, the main temperature over Europe is projected to continue to increase due to anthropogenic global warming [25]. Such an assessment is mainly based on the results of the different climate projections included in the fifth coupled model intercomparison project (CMIP5) [26], for different future emission scenarios, i.e., the representative concentration pathways (RCPs) [27,28]. Depending on the region, future warmer conditions may represent either an advantageous opportunity or a threat [29] by moving away from or by approaching the optimal climatic conditions for a given grapevine variety. This has the potential to overturn the geography of wine production by the end of the 21st century as suggested by many studies, e.g., [30–33], which prefigure a loss of suitability over the major present-day wine-producing areas and the establishment of new vineyards at higher latitudes or altitudes, however the extent of these changes is under debate [33,34]. These assessments have mainly been carried out by taking into account an ensemble of several climate model projections, e.g., the CMIP5. Yet, each model differs from the others due to different model parameterizations and numerical methods, defining a broad spectrum of possible climate projections. Their distribution states the inter-model uncertainty, while their ensemble mean is considered as the most reliable result. Indeed, comparisons between historical simulations and observational data demonstrated that multi-model mean generally outperforms most of, if not all, individual models [35,36]. This is likely because systematic biases intrinsically affecting individual models are, at least partly, cancelled by the averaging procedure [37]. This procedure, however, also cancels the internal variability out and all the large climatic oscillations reproduced by any individual model. Furthermore, an un-weighted multi-model mean tends to indiscriminately under-rate the probability of events that are physically plausible but scarcely reproduced by models due to their biases. For this reason, new methods are being developed to characterize the model response in relation to some emergent constraints [38], and to weight models according to their reliability for the simulation of a given phenomenon [39]. This approach eventually restricts the broad range of possible climate change scenarios and allows a better characterization of the uncertainty by dividing the models in different clusters depending on their response and on their reliability. Moreover, clustering enables the extraction of one or more model projections from the different subsets that can serve as case studies to analyze specific potential climate change scenarios and their impacts.

A similar approach has been adopted to analyze the North Atlantic temperature projections in the 40 CMIP5 models, which are characterized by a large uncertainty [40]. Sgubin et al. (2017) [41] found a strict link between the simulated temperature and the dynamical response of the subpolar gyre (SPG) oceanic convection, a key process for the heat exchange between the deep ocean and the atmosphere. Depending on the fate of the SPG convection in the projections, indeed, they identified three main cluster of models. Two models showed a large-scale Atlantic meridional overturning circulation (AMOC) disruption, provoking a gradual but strong temperature decrease all over the northern hemisphere, with peaks up to 4 °C in 50 years over Europe. Seven models exhibited an abrupt local collapse of the oceanic vertical convection in the SPG region, with temperature evolution characterized by a long-time increasing trend suddenly interrupted by single or multiple rapid drops, up to 3 °C over 10 years. The rest of the models, i.e., 31, did not show any abrupt change in the SPG convection, and were characterized by a continuous warming trend over the North Atlantic. Sgubin et al. (2017) [41] also argued that an assessment based only on an unweighted multi-model mean underestimates the occurrence of a SPG convection collapse, since the likelihood for such an event is enhanced if the model's reliability is accounted for. When considering only the most realistic models in simulating the present-day SPG ocean stratification, which has been shown to be an emergent constraint, the chance of an abrupt cooling event is almost as likely as a continuous warming trend, while the chance of a complete AMOC collapse is negligible. These findings highlight the necessity of specific impact analyses accounting for a scenario characterized by a SPG convection collapse. This is notably important for impact analyses over Europe, whose temperature changes are strictly

connected to those in the North Atlantic Ocean [42]. Sudden temperature drops over the North Atlantic have actually already been reported around 1970 [43], yet an analysis on their impacts on grapevine production in Europe is missing.

Under these premises, the aim of the present study is to investigate the implications of potential large temperature variations over Europe on viticulture practices at regional scale. For this purpose, we analyze different downscaled projections provided by the EURO-CORDEX exercise [44,45], and we mainly focus on the CSIRO-Mk6-3-0 model, which belongs to that cluster of CMIP5 models exhibiting a SPG convection collapse during the 21st century [41]. We present results for the RCP4.5 scenario, whose level of global warming is the closest to the 2 °C limit, a threshold often proposed as a potentially safe upper bound on global warming. Our choice, however, was also dictated by the limited number of downscaled projections simulating a SPG convection collapse within the EURO-CORDEX database. After a dynamical downscaling, the projected temperature data are used to force a hierarchy of phenological models simulating the main developmental stages of the grapevine. Their future evolution defines the climatic suitability for premium wine production. Current and new potential suitable winemaking areas are evaluated under the climate scenario prefiguring a SPG convection collapse and compared with the results shown by the ensemble mean of CMIP5 models. This comparison clearly marks the different impacts on viticulture coming out from different clusters of models, which should be carefully accounted for adaptation management.

## 2. Methods and Material

The methodology on which the present work is based can essentially be summarized in four main points, which also contain information about the material adopted:

- Simulation of coarse-resolution future climate by means of 7 CMIP5 general circulation models (GCM) under the RCP4.5 scenario.
- Dynamical downscaling over Europe according to the RCA4-SMHI model of 7 coarse-resolution GCMs.
- Coupling of the downscaled air temperature projections with 3 phenological models for the main developmental stages of the grapevine.
- Definition of climatic suitability for premium wine production based on estimated maturity dates.

### 2.1. General Circulation Models: the CMIP5 Simulations

Climatic projections are based on simulations of all the GCM participating to the CMIP5 project [26], which provides a standard protocol of daily data from the end of the pre-industrial era (historical simulations) to 2100 (future projections). The historical simulations run from 1850 to 2006, and the external boundary conditions consist of a prescribed radiative forcing representing all the known aerosol and greenhouse gases concentrations in the atmosphere estimated from observational data. The initial conditions are those obtained from the O (1000)-year control simulations based on stationary climatological forcing. The future projections start in 2006 and are forced by a common pattern of greenhouse gas concentration trajectories until 2100 describing different possible emission scenarios, i.e., the RCP scenarios [27,28]. Here, results from the RCP4.5. scenario [46] are analyzed, which prefigures a stabilization of radiative forcing at 4.5 W m$^{-2}$ by the end of the century. However, GCMs run at coarse spatial resolution, i.e., O (100) km, thus describing only large-scale processes and limiting impact analyses to global and continental scales.

### 2.2. Dynamical Downscaling with a Regional Circulation Model

For assessments at the regional scale, higher-resolution climate projections are required. For this scope, we use the EURO-CORDEX data (http://www.euro-cordex.net) [44,45], which provides CMIP5 climate projections at finer spatial grid, i.e., O(10) km, over Europe. These data are obtained by means of dynamical downscaling, a method consisting of running a regional circulation model

(RCM) starting from the GCM outputs over a limited area of the globe. The EURO-CORDEX data initiative offers an unprecedented number of simulations centered over Europe, thus constituting the benchmark dataset for future climate impact assessments. The whole dataset derives from 10 RCMs and 14 CMIP5 GCMs for the different RCP scenarios (updated on 2018) and is available at horizontal resolutions 0.44° (~50 km, EUR-44) and 0.11° (~12 km, EUR-11). Here, the outputs from the EUR-44 Rossby Centre regional atmospheric climate model (RCA4) [47,48] nested inside 7 different GCMs for the RCP4.5 scenario, i.e., CanESM2, CNRM-CM5, CSIRO-Mk3-6-0, GFDL-ESM2M, HadGEM2-ES, IPSL-CM5A-MR, MPI-ESM-LR, are analyzed. Furthermore, model outputs have been adjusted in order to ensure a statistical conformity between observational data and historical simulation. Our bias correction consists of aligning both mean and standard deviation of the model daily outputs to those calculated from WATCH observational data [49]. Such an adjustment has been carried out separately for each single month.

## 2.3. Phenological Models

Downscaled temperature projections over Europe have been successively used to force 3 different phenological models simulating the day of the year of occurrence for the main developmental stages of the grapevine. We carried out simulations for 4 different grapevine varieties, representative of different heat requirements for ripening [50,51], i.e., Chardonnay for early ripening variety, Syrah for middle ripening variety, and Cabernet Sauvignon and Grenache for late ripening varieties.

The phenological models used here assume that each developmental stage is exclusively induced by a sequence of certain temperature conditions. According to this approach, the day of occurrence of a given phenological stage $t_p$ coincides with the fulfilment of a critical temperature forcing $F^*$ formalized in terms of the cumulative daily forcing units $F_u$ after a certain starting day $t_0$:

$$t_p : \sum_{t_0}^{t_p} F_u = F^* \tag{1}$$

Depending on the different formulations of the function $F_u$ and on the different assumptions for $t_0$, three different phenological models have been here adopted: (i) A linear non-sequential model, (ii) a linear sequential model, and (iii) a non-linear sequential model.

The linear non-sequential model is a thermal time model [52], also known as a growing degree days (GDD) model [53], based on the cumulative heat forcing. In such a formulation, the forcing unit $F_u$ is a linear growing function of the daily mean temperature $T$, when this latter is greater than a base temperature $T_b$:

$$F_u = \text{GDD}_{T_b} = \begin{cases} 0 \text{ if } T \leq T_b \\ T - T_b \text{ if } T > T_b \end{cases} \tag{2}$$

Moreover, the thermal summation is calculated from a constant starting time $t_0$, meaning that each developmental stage is independent of the previous one. The budburst is based on a GDD model with a base temperature $T_b = 10$, and a fixed starting time $t_0 = 1$ January. Its formulation, parameterization, and validation have been provided in [54] by using a collection of 616 budburst measurements for 10 different grapevine varieties. The flowering and veraison have been calculated according to the grapevine flowering veraison model (GFV) [55], which is also based on a GDD model (Equation (2)). The daily sum of the forcing unit starts at $t_0 = 1$ March, i.e., the 60th day of year (DOY), and $T_b$ has been set at 0 °C. Its calibration and validation are based on a database corresponding to 81 varieties, 2278 flowering observations, and 2088 veraison observations, spanning from 1960 to 2007 and from 123 different locations over Europe. The maturity day has been instead assumed as occurring $k$ days after the simulated day of veraison, where the constant $k$ has been calculated as the average veraison-to-maturity period from more than 500 historical observations for the different grapevine varieties.

The linear sequential model is also based on a linear relation between daily temperature and forcing (chilling) unit, but the starting time of the sum of each phenological phase (Equation (1)) is not fixed a priori but depends on the previous phenological stages. The budburst model is based on the BRIN model [54], which includes dormancy and post-dormancy sub-models, thus allowing the simulation of the dormancy break from which the summation of $F_u$ starts. The dormancy break $t_{db}$ sub-model is based on the accumulation of chilling unit $C_u$ until a critical value $F^*$ is reached:

$$t_{db} : \sum_{t_0}^{t_{db}} C_u = C^* \tag{3}$$

with $C_u$ formalized according with the $Q_{10}$ Bidabe's formula [56]:

$$F_u = Q_{10}^{\frac{-T_{\max}}{10}} + Q_{10}^{\frac{-T_{\min}}{10}} \tag{4}$$

where $T_{min}$ and $T_{max}$ are, respectively, the minimal and the maximal daily temperatures, $Q_{10}$ is an a-dimensional constant set at 2.17. The post-dormancy calculation follows the method of Richardson [57], which is based on the growing degree hours (GDH) cumulated over a day, so that the forcing unit $F_u$ in Equation (2) is here approximated as:

$$F_u = \sum_{h=1}^{24} \text{GDH}_{T_b} \approx \begin{cases} \frac{T_{\max}+T_{\min}}{2} - T_b \text{ if } \frac{T_{\max}+T_{\min}}{2} \leq T_B \\ T_B - T_b \text{ if } \frac{T_{\max}+T_{\min}}{2} > T_B \end{cases} \tag{5}$$

where $T_B$ is the upper base temperature, here set at 25 °C, beyond which development rate becomes constant [58], while the (lower) base temperature $T_b$ is 5 °C. The parameterization and validation of the BRIN model are based on a database corresponding to 10 grapevine varieties and 616 budburst observations [54]. The flowering and the veraison are based on a $GDD_{10}$ (Equation (2)), whose summation start when budburst is accomplished. The day of maturity is calculated according to a $GDD_{10}$ model and a starting time coinciding with the budburst occurrence. The parameterization of the different critic temperature accumulation $F^*$ and their validation have been provided in [59].

The non-linear sequential model is based on a curvilinear response to the temperature for the calculation of flowering, veraison, and maturity. As for the linear sequential model, the budburst model is also based on the BRIN model [54], which represents the only linear component of this model. The following phenological phases are instead based on a non-linear formulation of the forcing unit $F_u$, which is determined by three cardinal temperatures, i.e., a base temperature $T_b$, a limit temperature $T_{lim}$, and an optimal temperature $T_{opt}$ [60]:

$$F_u = \begin{cases} \frac{2(T-T_b)^\alpha (T_{opt}-T_b)^\alpha - (T-T_b)^{2\alpha}}{(T_{opt}-T_b)^{2\alpha}} \text{ if } T_b \leq T \leq T_B \\ 0 \text{ if } T < T_b \text{ or } T > T_B \end{cases} \tag{6}$$

where

$$\alpha = \frac{\ln 2}{\ln\left(\frac{T_B - T_b}{T_{opt} - T_b}\right)} \tag{7}$$

Its curvilinear structure allows it to consider the effects of high temperatures on development slowdown [61]. Cardinal temperatures $T_b$ and $T_{lim}$ have been fixed, respectively, to 0 °C and 40 °C, while optimal temperature $T_{opt}$ and the critical forcing $F^*$ are obtained from [62]. The values of all the parameters for the different phenological models are summarized in Table 1.

**Table 1.** Values of parameters and calibration of the different phenological models for the different varieties.

| | | Linear Non-Sequential | | | Linear Sequential | | | | Curvilinear Sequential | | | | |
|---|---|---|---|---|---|---|---|---|---|---|---|---|---|
| | | $t_0$ | $T_b$ (°C) | F* | $t_0$ | $T_b$ (°C) | $T_B$ (°C) | F* | $t_0$ | $T_b$ (°C) | $T_B$ (°C) | $t_{opt}$ (°C) | F* |
| *Budburst* | Chardonnay Syrah Cabernet S. Grenache | DOY = 1 | 5 | 220.1 265.3 318.6 321.3 | $t_{DB}$ | 5 | 30 | 6577 7819 / 9174 | $t_{DB}$ | 5 | 30 | / / / / | 6577 7819 9169 / |
| *Flowering* | Chardonnay Syrah Cabernet S. Grenache | DOY = 60 | 0 | 1217 1279 1299 1277 | $t_{BUD}$ | 10 | 40 | 253.9 313.3 / 327.7 | $t_{BUD}$ | 0 | 40 | 30.3 32.0 30.2 / | 18.8 12.5 20.3 / |
| *Veraison* | Chardonnay Syrah Cabernet S. Grenache | DOY = 60 | 0 | 2547 2601 2689 2761 | $t_{BUD}$ | 10 | 40 | 951 1012 / 1148 | $t_{FLO}$ | 0 | 40 | 24.3 27.0 24.3 / | 56.2 52.8 63.0 / |
| *Maturity* | Chardonnay Syrah Cabernet S. Grenache | DOY = $t_{VER}$ | / | K = 41 K = 46 K = 52 K = 51 | $t_{BUD}$ | 10 | 40 | 1675 1685 / 1926 | $t_{VER}$ | 0 | 40 | 24.3 27.0 24.3 / | 46.0 43.2 51.5 / |

## 2.4. Definition of Climatic Suitability for the Different Grapevine Varieties

We introduce the concept of climatic suitability for premium wine production by means of the definition of an optimal temporal window for the maturity day. Here, we assume this to range between 10 September and 20 October, similarly to the time interval proposed in [2], in which, however, the upper limit was fixed to 10 October. According to this assumption, hence, the climatic conditions are favorable for the production of high-quality wine if the maturity day falls within this specific period of the year. The definition of climatic suitability intrinsically states the stability of the traditional vineyards under climate change as well the opportunity for new regions to become appropriate for high-quality production. However, it is important to stress that our definition of suitability only accounts for the thermal conditions for ripening, yet other parameters can be also important.

## 3. Results

### 3.1. Uncertainty in Climate Projections and Model Clustering

As shown in the multi-model analysis in [41], different behaviors of the oceanic circulation in the North Atlantic SPG led to divergent temperature projections over that region, which defined three main distinct clusters of models. Their characterization implies three different temperature trends over the SPG as well as the occurrence or not of an abrupt cooling. In order to evaluate if these different temperature behaviors over the SPG also propagates in the surrounding regions and penetrate over the continents, Figure 1 shows the maximum 10-year temperature drop throughout the 21st century over Europe, against the 100-year temperature trend for each available CMIP5 projection. Such a 2D diagram groups model projections according to their inter-decadal variability and their long-term temperature change. The distribution of the single models and their ensemble means and spread are shown for the 37 non-downscaled RCP4.5 projections (Figure 1a) and for the 7 projections downscaled with the SMHI-RCA4 regional model (Figure 1b).

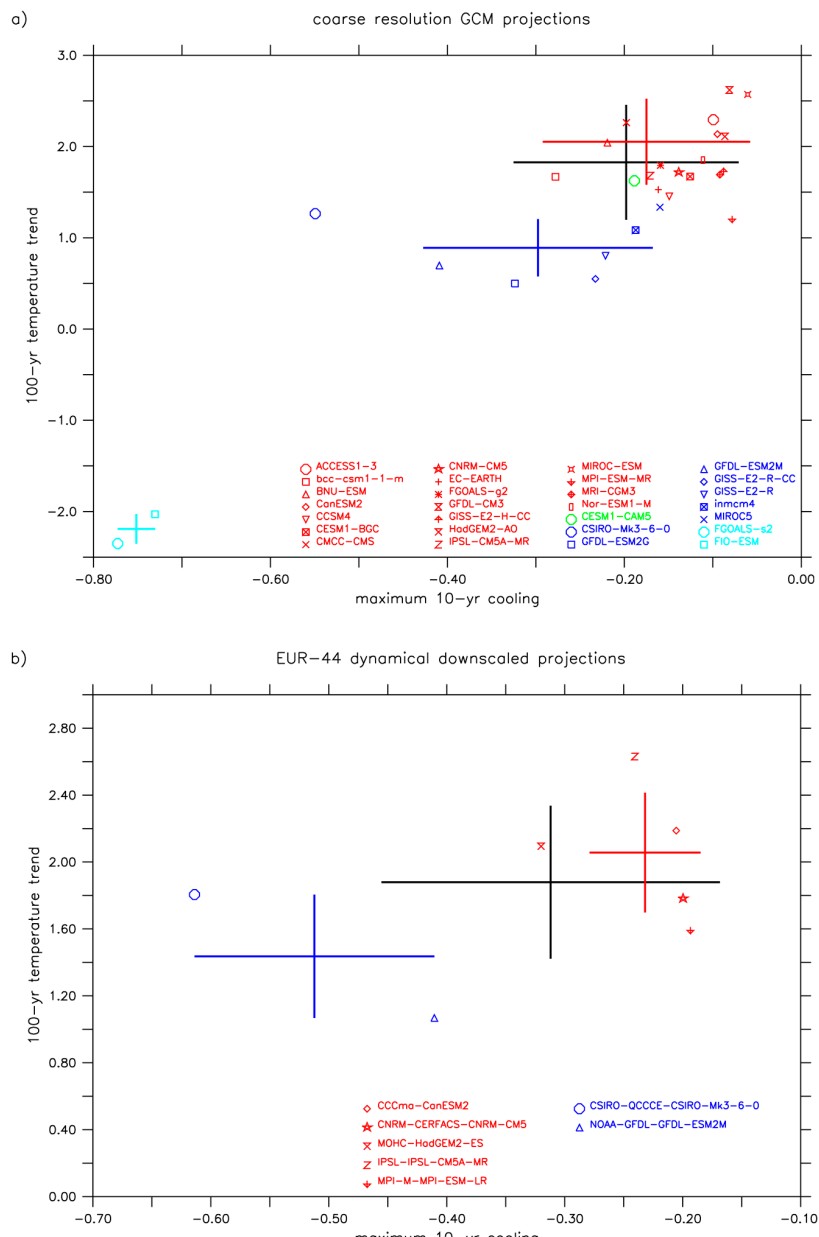

**Figure 1.** Scatterplot of the simulated 100-year temperature trend (in °C) versus the maximum 10-year cooling event (in °C) over western Europe for (**a**) the 37 coarse-resolution fifth coupled model intercomparison project (CMIP5) projections and (**b**) the 7 dynamical downscaled projections for the representative concentration pathways (RCP)4.5 scenario. Different colors follow the model clustering proposed in Sgubin et al., 2017, which groups projections not showing any abrupt change in the subpolar gyre (SPG) (red), projections producing a SPG convection collapse (blue), and projections simulating an Atlantic meridional overturning circulation (AMOC) collapse within 2100 (cyan). The green point in the upper panel corresponds to CESM1-CAM5, which is a model showing a SPG convection collapse, but only for RCP8.5. Crossing lines individuate the mean and the standard deviation of each subset of models, with the black lines corresponding to the ensemble of all the models. For multiple models developed at the same institute, we displayed just one point for a matter of readability in the diagram, e.g., among the different models developed by the Institut Pierre Simon Laplace (IPSL), we just displayed results of IPSL-CM5A-MR. However, all the projections have been considered in the calculation of the different clusters' ensemble mean and spread.

When all the 37 GCMs (7 downscaled projections) are considered, the ensemble mean temperature trend over Europe is $1.83 \pm 0.63$ ($1.88 \pm 0.46$) °C/century while the maximum 10-year cooling is, on average, $-0.20 \pm 0.13$ ($-0.31 \pm 0.14$) °C/decade. The distribution of the single-model results on this diagram evidences that the clustering proposed in [41] for the characterization of the sub-polar North Atlantic temperature response also subsists when Europe is analyzed. Indeed, three main distinct subsets of models can clearly still be identified for the temperature response over the continent. In Figure 1a, models simulating an AMOC disruption (cyan) are characterized by a 100-year cooling trend ($-2.2 \pm 0.16$ °C/century) and by a maximum 10-year cooling of $-0.75 \pm 0.02$ °C/decade. The models projecting a SPG convection collapse (blue) are all characterized by a smaller than average warming trend, i.e., $0.89 \pm 0.31$ °C/century, and/or by a stronger than average 10-year cooling events, i.e., $-0.30 \pm 0.13$ °C/decade. The rest of the models (red) exhibits higher temperature trends over Europe, i.e., $1.89 \pm 0.37$ °C/century as well as slighter or almost null cooling episodes, $-0.13 \pm 0.06$ °C/decade. These models, being the majority, strongly influence the ensemble mean of all the models. A similar pattern is qualitatively valid when considering the responses of the seven downscaled projections (Figure 1b), which, however, just include two models showing a SPG convection collapse and five models not showing any abrupt change in the North Atlantic. The latter shows a mean temperature trend $2.06 \pm 0.36$ °C/century and a maximum 10-year cooling of $-0.23 \pm 0.05$ °C/decade, while the former is characterized by a subdued warming trend, i.e., $1.43 \pm 0.37$ °C/century) and by larger 10-year temperature oscillations, i.e., $-0.51 \pm 0.10$ °C/decade.

### 3.2. The Spatial and Temporal Features of the Cold Waves over Europe

Since the number of models showing an abrupt decadal-scale cooling is much lower than the models not showing any abrupt cooling, an assessment based on the ensemble mean of all the CMIP5 models covers, to some extent, the possibility of a SPG convection collapse and its associated temperature oscillations affecting the European climate. However, the likelihood of such an event has been actually assessed to be higher than what the unweighted CMIP5 multi-model ensemble mean shows [41]. In order to take into account such a possibility, we therefore differentiate the impact analysis by separating the results of one of the projections reproducing a SPG convection collapse from the results evidenced by the multi-model mean classical procedure. In Figure 2, the temperature evolutions simulated by the CSIRO-Mk3-6-0 model for different European regions are displayed and compared with the ensemble mean trend of the 37 projections. The response to the RCP4.5 emission scenario prefigured for the 21st century is characterized by a long-term warming trend all over the Europe, in line with all the models here analyzed (Figure 2). In addition, the CSIRO-Mk3-6-0 projection is also characterized by a strong inter-decadal variability, with multiple cooling events that interrupt, for a certain period, the long-term warming trend. It is possible, indeed, to identify three main decadal cold waves along the 21st century, which make this model the one featuring the largest multi-decadal variability over Europe among the downscaled projections (Figure 1b). It is worth emphasizing that these simulated cold waves over Europe occur in concomitance with an abrupt reduction of the oceanic convective activity in the SPG, which prevents the local heat exchange from the deep ocean to the surface normally occurring in winter, and cause temperature to drop locally, despite the global warming signal (see Figure S1 in Supplementary Materials). Although the identification of the driver of the cold wave is not the aim of this study, the fact that the three decadal cooling events over Europe simulated by the CSIRO-Mk3-6-0 model coincide with abrupt reductions of the convection activity in the SPG reinforces the hypothesis of a strict connection between ocean circulation changes in the North Atlantic and rapid climate oscillations over Europe.

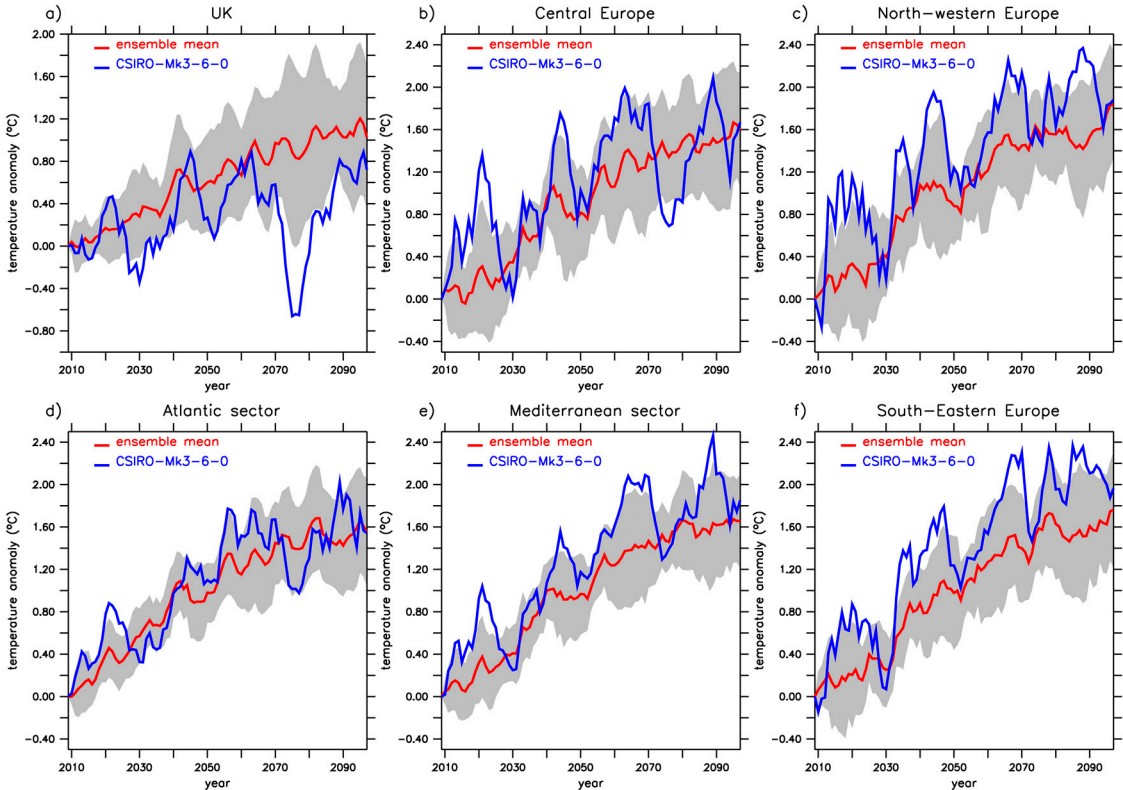

**Figure 2.** Evolution of the 2 m air temperature for the 21st century RCP4.5 projections over different regions of Europe: (**a**) The U.K., (**b**) Central Europe, (**c**) North-eastern Europe, (**d**) Atlantic sector, (**e**) Mediterranean sector, and (**f**) South-eastern Europe. Blue lines indicate the temperature evolution simulated with the CSIRO-Mk3-6-0 model. Red lines indicate the seven downscaled models ensemble mean temperature evolution, which is embedded in the grey portion indicating the inter-model spread, i.e., the standard deviation.

Despite a long-term warming trend in line with the other models, the temperature evolution simulated by the CSIRO-Mk3-6-0 model largely deviates from the continuous and gradual warming pattern characterizing the CMIP5 ensemble-mean. The cooling event simulated around 2075 even exceeds, by two times, the standard deviation of ensemble mean temperature over the U.K. and continental part of Europe, thus representing an outstanding case of study for the analysis of the impacts of large temperature oscillations over Europe. We, therefore, focus on this specific cooling event simulated by the CSIRO-Mk3-6-0 model, with the intent of characterizing its possible effects on climate over Europe, which can be overlooked by a multi-model ensemble procedure. In Figure 3, we show the anomaly of temperature over Europe during the 10-year cooling event (2069–2078) with respect to the previous 10 years (2059–2068). The pattern of such a long-lasting simulated "cold blob" appears to form in winter over the northern North Atlantic and to propagate towards Europe, losing its intensity in south-east direction, towards the Mediterranean Sea and the Black Sea, where it extinguishes. At the annual time-scale, the temperature drop mainly involves the U.K. and the continental region of Europe (Figure 3a), including areas indicated as the most suitable for new grapevine plantations. The intensity of the 10-year cooling is larger during the winter (Figure 3b), while it weakens with the following seasons (Figure 3c–e). In winter, the cold blob that formed over the North Atlantic Ocean mainly runs over the U.K. and most of the central regions of Europe, from France to Poland, where winter temperature in the decade 2069–2078 are, on average, 2 °C colder than in the previous decade. In spring, most of the Europe is touched by the cold wave, whose core is centered more south-eastern with respect to winter, notably involving the regions surrounding the Alps and the Balkans. The effect

of the cold blob starts to fade in summer, while in autumn it appears to vanish over the continent, while still affecting the temperature of the North Atlantic Ocean.

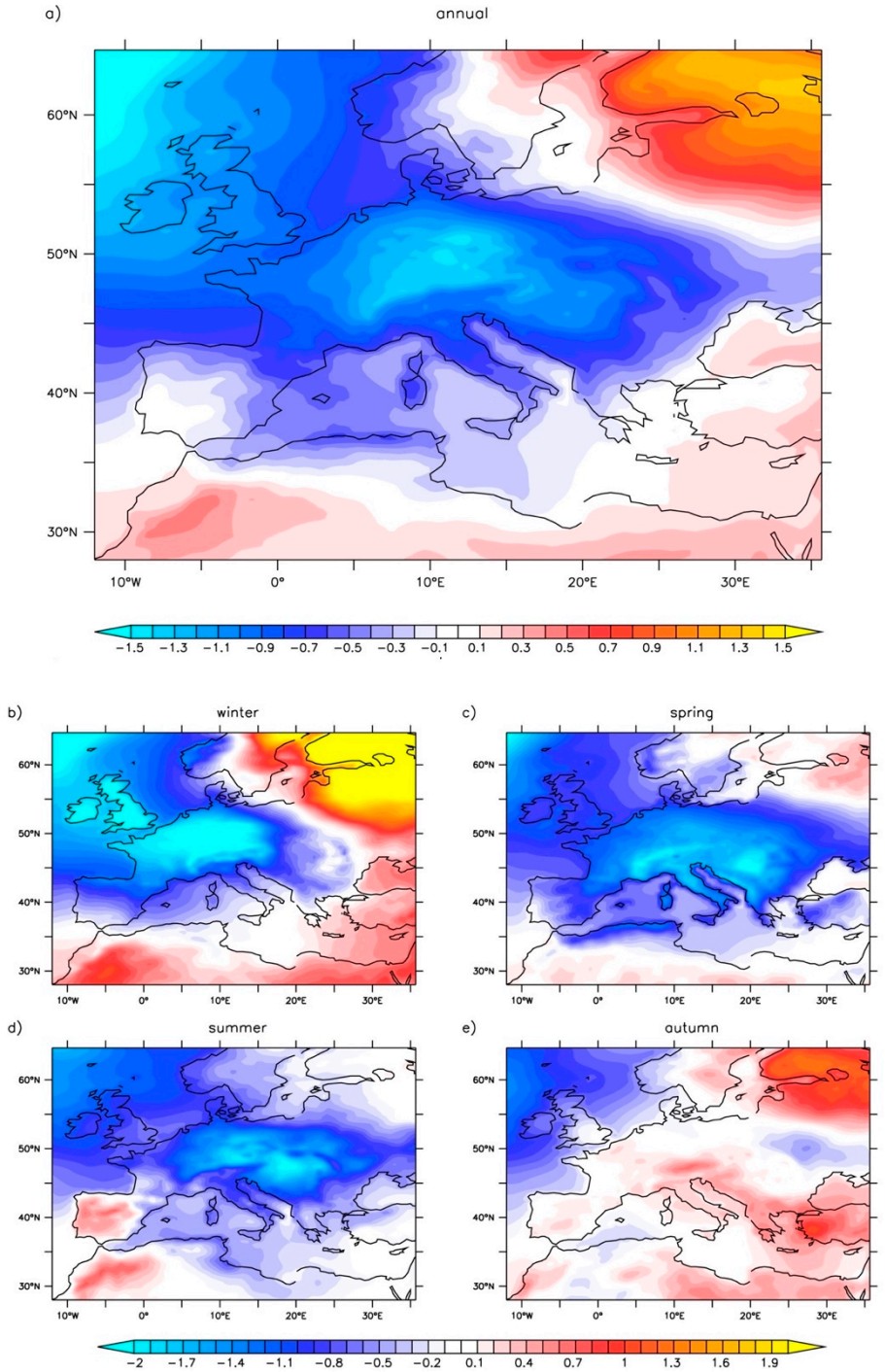

**Figure 3.** Pattern of the mean air temperature anomaly (in °C) between the decade 2069–2078 and the decade 2059–2068 as simulated by the downscaled CSIRO-Mk3-6-0 model for different periods: (**a**) Annual, (**b**) winter, (**c**) spring, (**d**) summer, and (**e**) autumn.

### 3.3. The Effect of the Rapid Cooling on Phenology of the Grapevine

The distinct seasonal responses imply that the different phenological phases of the grapevine are not affected in the same way during the occurrence of the cold wave. In Figure 4, we display the anomaly between the decades 2069–2078 and 2059–2068 of the simulated occurrence of the main

phenological stages for Chardonnay (an early variety; for the other grapevine varieties, see Figures S2–S4). Such an anomaly is carried out by using the ensemble of the three phenological models used here, while details related to each of the single phenological models are illustrated in Figures S5–S7 of Supplementary Material. All the growing phases of the grapevine appear to occur later (positive anomaly) in the decade 2069–2078 all over Europe, as a consequence of the large cooling with respect to the previous decade.

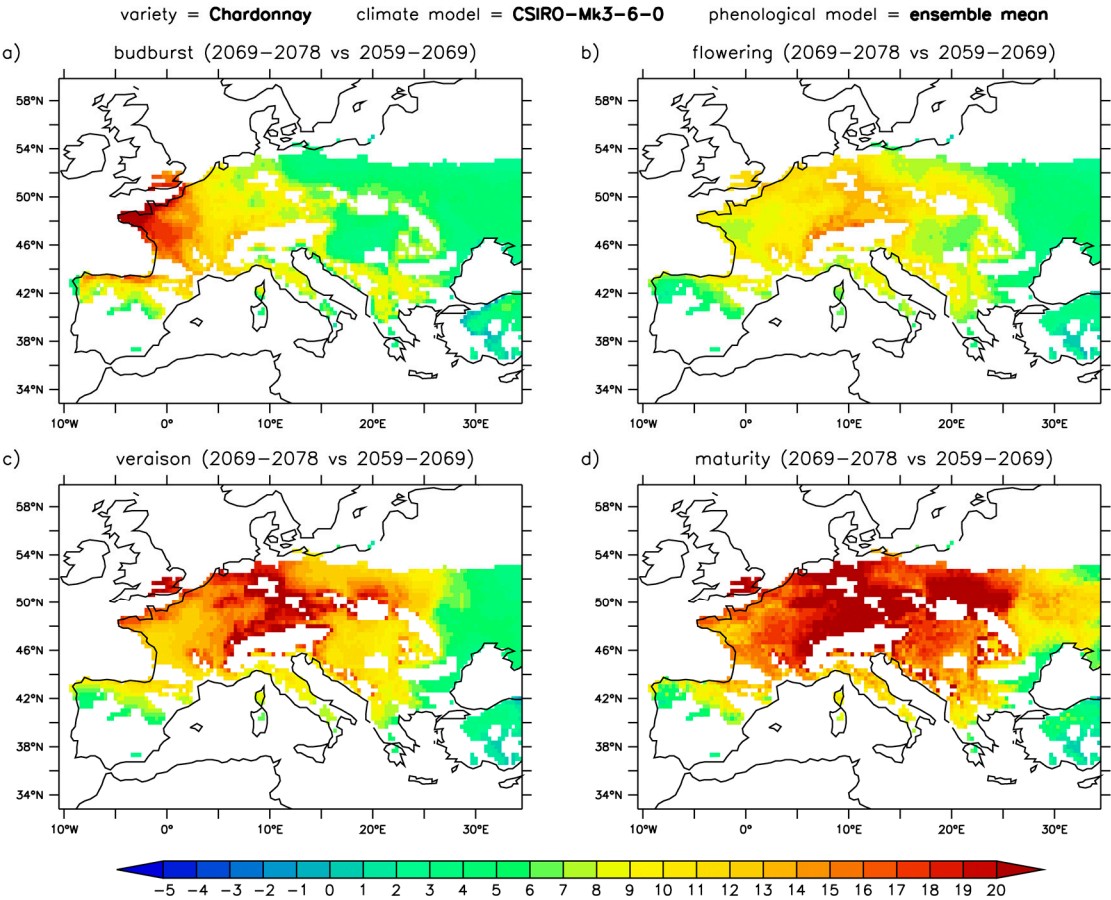

**Figure 4.** Pattern of the anomaly (in days) of the occurrence of the phenological stages, i.e., (**a**) budburst, (**b**) flowering, (**c**) veraison, and (**d**) maturity, for the Chardonnay variety between the decade 2069–2078 and the decade 2059–2068 as simulated by the downscaled CSIRO-Mk3-6-0 model. Results are based on the ensemble mean of the three phenological models here adopted.

The average delay of the budburst is about 10 days over Europe, with peaks of delay mainly concentrated over the Atlantic sector of France and the UK, where the anomaly is around 20 days (Figure 4a). In general, late budburst notably involves the western part of Europe, while in the eastern part of Europe, it is limited to a few days. The mean delay of the flowering is also about 10 days, but much more uniform in space, with peaks between 10 and 15 days, mainly located in the central part of Europe (Figure 4b). The cumulated lags led to large veraison anomalies, up to 15–20 days over most of the central part of Europe. This eventually led to maturity dates strongly delayed in the period 2069–2078 compared to the previous decade, i.e., on average by 15 days over Europe, with peaks of almost one month over the central part of Europe. This represents a significant delay, notably if we consider that such anomalies take place in less than 10 years. These abrupt changes may have high repercussions on the production and quality of wine.

### 3.4. The Climatic Suitability for Premium Wine Production During the Cold-Wave Events

A main threat for winemakers in the context of climate change concerns the conservation of *terroir* characteristics, and if the future climatic conditions will still be favorable for the production of high-quality wine. As illustrative examples, in Figure 5 we show the evolution of the main phenological phases for typical varieties in four different renowned wine production regions, according with the downscaled CSIRO-Mk3-6-0 projection.

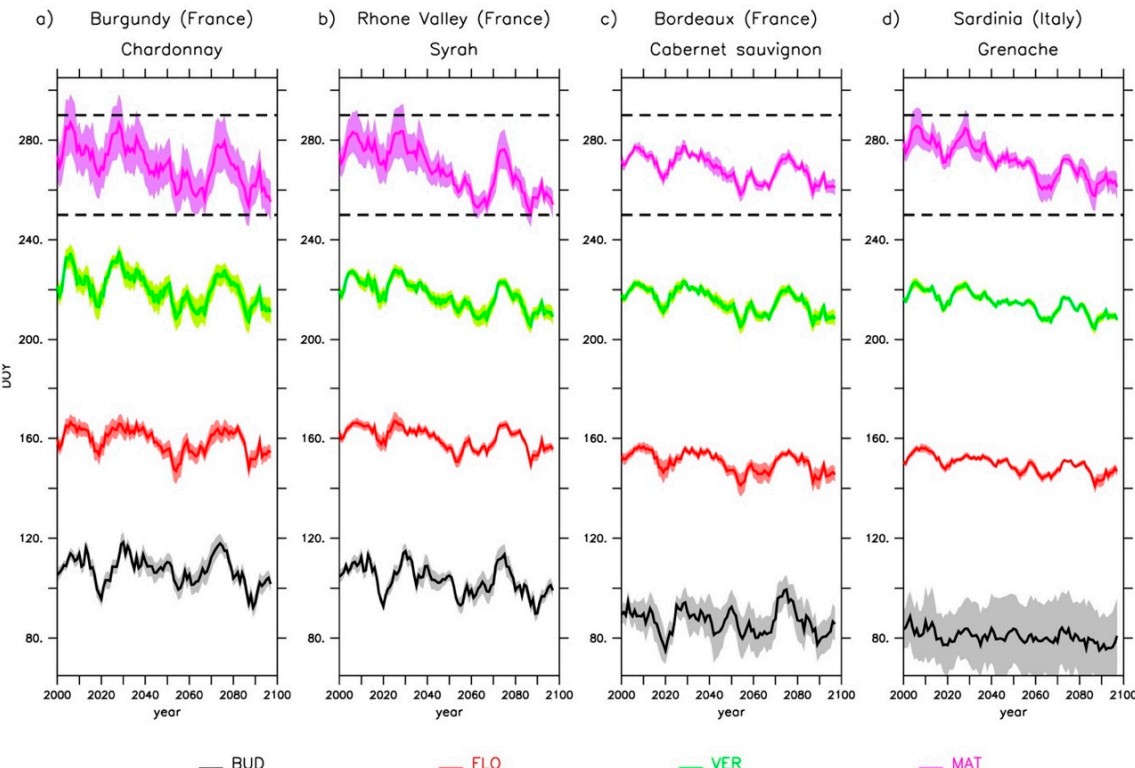

**Figure 5.** Evolution of day main phenological phases (in day of year (DOY)) of typical grapevine varieties in four selected renowned wine-making regions of Europe: (**a**) Chardonnay in Burgundy, (**b**) Syrah in Rhone Valley, (**c**) Cabernet Sauvignon in Bordeaux, and (**d**) Grenache in Sardinia, as simulated by the downscaled CSIRO-Mk3-6-0 model. Results are based on the ensemble mean of the three phenological models here adopted. Black lines indicate the budburst, red lines the flowering, green lines the veraison, and violet lines the maturity evolution, while their shaded intervals represent the two-sigma spread between the three phenological models. Dashed lines indicate the limits for maturity day within which climate conditions are suitable for premium wine production according to the definition in Methods and Materials.

At the centennial time-scale, the results evidence a widespread trend to an anticipation of all the phenological stages in all the four selected checkpoints grapevine varieties. This is the direct effect of the general warmer conditions throughout the 21st century. The trend towards earlier stages appears slight or even null for budburst, while it is more pronounced for successive stages, in particular for maturity. Superimposed on this trend, each phenological stages is characterized by a significant inter-decadal variability, mainly associated with the multiple temperature drops over Europe evidenced by the CSIRO-Mk3-6-0 model. This implies large oscillations in the characteristics of the fruit composition and therefore in the vintage rating, likely affecting the economical resilience of wine businesses. Nevertheless, by using the definition of climatic suitability for high-quality production we introduced in the section Methods and Materials, Figure 5 also shows that the maturity, despite its long-term trend towards precocity and its strong inter-decadal variability, always falls in the optimal range for premium wine production in the four illustrative examples here analyzed. We can, therefore, claim that

these typical varieties in these traditional sites appear to be resilient according with the CSIRO-Mk3-6-0 model. The same feature is, to some extent, also valid when the ensemble mean is considered (see Figure S8). The simulated maturity day always falls within the range of optimal ripening. However, it approaches its lower limit, so that further warming during the 21st century due to a more severe emission scenario likely enshrines the necessity of varietal shifts.

Climate change may promote the settlement of new regions for high-quality wine production and force winemakers in the current vineyards to adapt to warmer conditions by replacing traditional varieties with later ripening varieties. In Figure 6, the same analysis as in Figure 5 was carried out but for potential new emerging wine regions (upper panels) and for traditional regions where we assumed hypothetical replantation with later ripening varieties (lower panels). The general pattern of the evolution of all the phenological phases is qualitatively similar to those shown in Figure 5. However, at beginning of the century, none of the maturity associated with the grapevine varieties selected for these regions falls within the optimal temporal window for high-quality wine production. This well-reflects the actual present-day suitable areas for the different grapevine varieties, whose northern limit generally does not exceed the 50° N parallel. As the temperature increase during the 21st century and the grapevine development becomes faster, maturity dates of the grapevine varieties selected for these six regions starts to fall within the suitable range for premium production. Therefore, climate change appears to be beneficial for plantations in cool climate regions as well as being compatible with variety replacements in the traditional regions. Nevertheless, during the abrupt cold event simulated in the decade 2069–2078, the optimal climatic conditions in these regions are not satisfied anymore, as they would produce too-late maturity dates. Depending on the specific region, the loss of climatic suitability appears to last from a few years to approximately 15 years, thus questioning the economic viability of those adaptation strategies presupposing northward varietal shifts [63], which are irreversible adaptations in the short-term. These results differ from those carried out by the multi-model ensemble mean (see Figure S9). Indeed, warming trend causes maturity dates to persistently fall within the range of suitability after a certain period, although at the beginning of the century, none of the selected regions are characterized by optimal climatic conditions for premium wine production. Overall, we can, thus, claim that while the long-term warming signal may represent an opportunity for new vineyards areas and may be compatible with plantations of later ripening varieties in traditional regions, the climate decadal variability represents a serious risk that could compromise the quality of wine production for a relatively long period and therefore the economic investments implied in these adaptation measures.

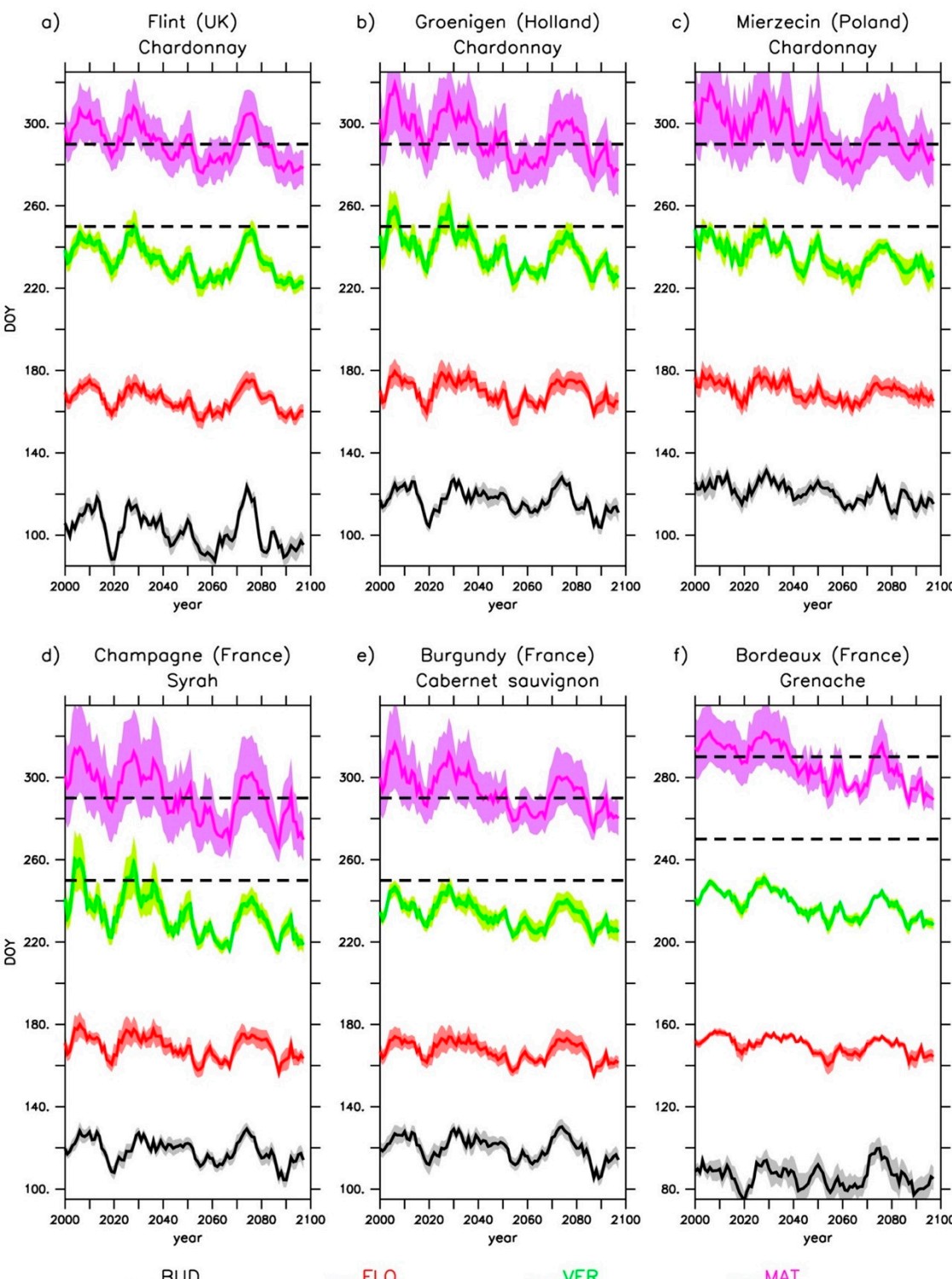

**Figure 6.** As Figure 5 but for different varieties and locations: (**a**) Chardonnay in UK, (**b**) Chardonnay in Holland, (**c**) Chardonnay in Poland, (**d**) Syrah in Champagne, (**e**) Cabernet Sauvignon in Burgundy, and (**f**) Grenache in Bordeaux.

Focusing on Chardonnay (see Figures S10–S12 for the other varieties), we summarize in Figure 7 the changes in climatic suitability for premium production over Europe during the 2069–2078 cold wave with respect to the previous decade. In this period, most of the supposed "new cool wine regions" [64,65] appears to lose their climatic suitability, from U.K. to Poland, as a consequence of the

rapid cooling. In contrast, the south-west of France, part of the Balkans, and some hilly regions in Spain and Italy would benefit from the temperature decrease by recovering their climatic suitability lost throughout the 21st century. This pattern is nearly specular to the changes in climatic suitability between the decade 2059–2068 (just before the occurrence of the cold event), and the present-day decade, i.e., 2010–2019: The effect of 50-year climate changes produces a shift of climatic suitability towards the north (Figure S13). This means that the occurrence of cold waves has the potential to abruptly cancel, at least for a decade, all the beneficial effects of warmer conditions accumulated since the beginning of the century, which are projected by the totality of the state-of-the-art climate models for the RCP4.5 scenario.

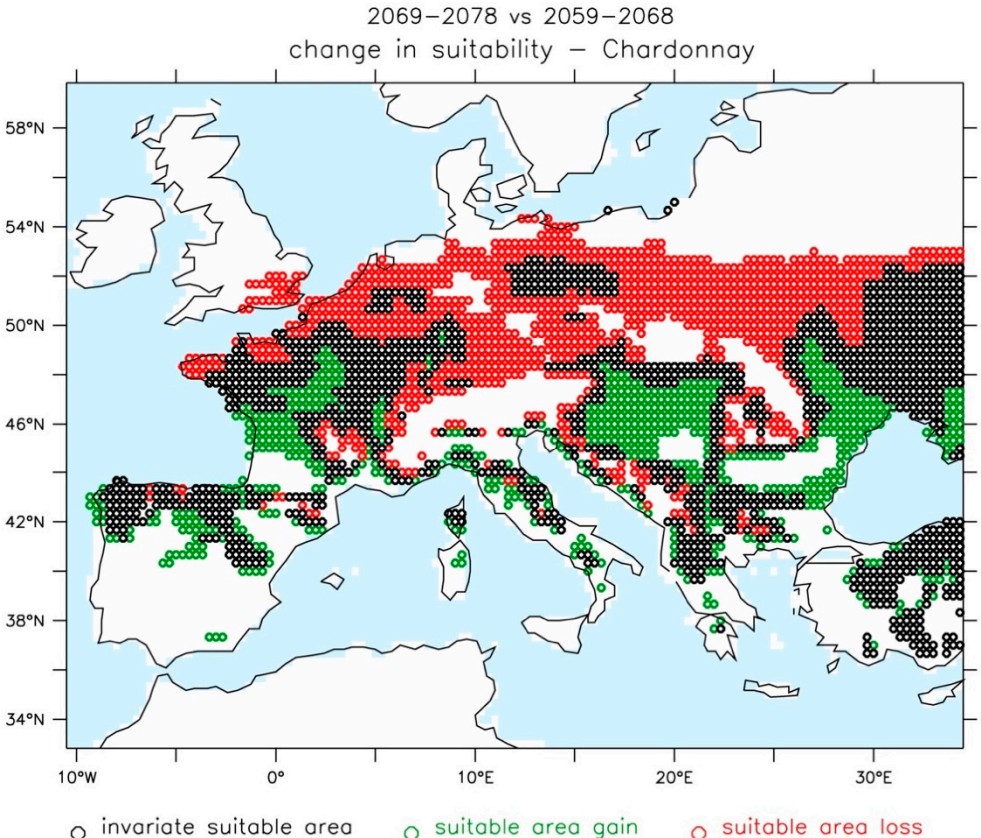

**Figure 7.** Anomaly in climatic suitability for premium wine production between the decade 2069–2078 and the decade 2059–2068 as simulated by the downscaled CSIRO-Mk3-6-0 model. Climatic suitability is based on the definition we introduced in Methods and Material. Over a decade, a location is assumed to be suitable when the climatic conditions are satisfied for at least 70% of the years. Results are based on the ensemble mean of the three different phenological models. Black circles indicate locations for which climatic conditions are suitable for both the 2059–2068 and the 2069–2078 periods. Green (red) circles indicate sites where climatic conditions are suitable for the decade 2069–2078 (2059–2078) but not for the decade 2059–2078 (2069–2078).

## 4. Discussion

The endorsement of measures of adaptation to climate change presupposes a deep investigation of both the beneficial and the deleterious potential return of all the proposed actions. This implies considering all the wide range of plausible future climatic changes at regional scale, in order to optimize the economic investments and to minimize the risks related to the occurrence of overlooked events. In this context, the assessment of a very likely long-term gradual temperature increase along the 21st century, which is based on the ensemble mean of the state-of-the-art climate projections, is orienting decision-makers and stakeholders to rethink the grapevine cultivation zoning, prefiguring a varietal

shift at higher latitudes and/or at higher altitudes. The latter are long-term adaptation measures that entail considerable investments and a non-immediate benefit. However, in the assessment of a gradual long-term warming, the effects of decadal variability are missing. In Europe, rapid temperature variations may occur as a side effect of the climate change, producing decadal-scale large local cooling events in a context of global warming [41], as shown, for instance, by the CSIRO-Mk3-6-0 model. The present study arises from the need of considering the possibility of these cold waves in an impact analysis for viticulture. We highlight that the occurrence of sudden cooling events over Europe has implications for grapevine growing that largely differ from those associated with a gradual warming. This finding may promote the application of our methodology to other relevant crops for Europe, e.g., wheat and maize.

Taking into account, at the same time, the possibility of a long-term continuous warming and the possibility of rapid cold events poses further challenges in the planning of proper measures of adaptation for wine-producing sectors in Europe, and in the analysis of their economic impact and sustainability. Indeed, beyond the effective costs that the expected increase of temperature will presuppose for preserving the wine production over Europe, our results suggest a further economic evaluation of the risks associated with an amplified decadal variability. This suggests the research of producing systems that are able to face, at the same time, the warming trend and the possibility of unprecedented large and rapid changes over Europe. Such an approach implies an accurate analysis of the sustainability of radical choices, thus promoting a more rational strategy of adaptation, e.g., a diversified variety relocation, which takes into account the risk of sudden temperature changes for each specific region. Also, this approach can partly resize the pursuit of new wine regions, notably in those regions appearing particularly sensitive to large inter-decadal variability. For example, according to our results, hypothetical new vineyard settlements would be more suitable in the south-eastern part of Europe rather than in its north-western part, since the latter would be primarily impacted by the occurrence of the cold waves originated in the North Atlantic. These considerations become notably relevant if one considers the large economic effort presupposing the potential settlement of new vineyards in cool climate regions.

Under the hypothesis of a stabilization of the greenhouse gas emissions within 2100 and a level of global warming limited to approximatively 2 °C, our results also suggest that varietal changes in traditional wine-producing regions do not appear strictly necessary. Indeed, over most of these regions, climate conditions appear to remain always suitable for typical grapevine varieties, both in the case of a gradual warming as evidenced by the model ensemble mean, and in the case of rapid decadal cold waves throughout the 21st century as shown by the CSIRO-Mk3-6-0 model. However, for more severe emission scenarios like RCP8.5, a northward varietal shift may appear as the most proper adaptation measure. Hence, the potential benefits of such a strategy appear particularly conditional on the capacity of mitigation of climate change. Moreover, the actual feasibility of varietal replacements is strictly dependent on regional *terroir* regulations, whose conventions would be arduous to change due to the cultural legacy at the base of local wine identities.

It is worth stressing that our definition of suitability for a given grapevine variety is exclusively based on thermal conditions for optimal maturity. It considers the cumulative temperature forcing during the growing season, meaning that the risk of extreme meteorological events like spring frosts [66], heat waves, and droughts, which have detrimental effects on suitability, are not taken into account. The definition of an indicator that also includes these factors is the subject of further studies.

Independently of the change in climatic suitability, the occurrence of abrupt temperature oscillations may imply rapid changes on the wine composition and organoleptic features, likely implying negative repercussions on the wine market [67]. In this regard, progresses in oenological and viticultural practices may subdue the negative effects of the short-term large climate variations by adjusting the year-to-year wine composition according with the needs [68]. In parallel with assessments of the long-term climate change, an optimal management of the vineyards is therefore conditional on the capacity of anticipating with accuracy the decadal climate variability and therefore

the occurrence of potentially forthcoming rapid climate changes. The recent advances in climate decadal predictions are promising in this sense, as a skilled forecast system can effectively support operational adaptation measures at the short-term time scale. For example, predicting the mean growing seasons for the following few years may promote strategic procedures of canopy management, aimed at optimizing leaf-area-to-fruit-weight ratio [69], and therefore the grape development and the wine composition. Also, an assessment of the risks associated with possible extreme decadal-scale events may prompt producers to deploy remedial infrastructures, e.g., wind machines, sprinkling water machines, gas-powered heaters, anti-hail nets, or to stipulate specific insurances, thus minimizing the possible economic damage of hail and frost events. The reliability of climate decadal predictions and their effective applicability in the context of adaptation strategies for European vineyards management will be the subject of future studies.

## 5. Conclusions

In this study, we analyzed, in detail, the impact of potential rapid temperature drops on viticulture over Europe, which have been previously demonstrated to be plausible events that can superimpose on the long-term warming trend along the 21st century [41]. We focused on the decadal-scale cold waves over Europe projected by the CSIRO-Mk3-6-0 model for the RCP4.5 scenario, and we compared their impact on grapevine growing with that resulting from the ensemble mean of seven climate projections producing a progressive warming signal. Our results evidenced that the occurrence of the cold waves yields significant changes in all the developmental stages of the grapevine, which would be able to overturn the long-term warming effect, at least for time steps of approximately a decade. During these cold events simulated in the future, climate conditions became rather similar to the present-day conditions in a very short period of time (a few years), thus rapidly cancelling out the previous warming that was gradually taking place since the beginning of the century. By defining the climatic suitability for premium wine production as those conditions satisfying the temperature requirements for the grapevine ripening to fall within a specific period of the year, we reported a potential loss of suitability during the occurrence of cold wave events over most of the central-western part of Europe. The same regions were those that became previously suitable, due to the simulated 21st century gradual warming. Our findings therefore disclosed the possibility that long-term adaptation measures like varietal northward shifts may not be the most appropriate in those regions potentially strongly hit by cold wave events. The most sensitive region includes the U.K. and different countries of central Europe (e.g., Holland, Germany, Poland), which have been often identified as new "cool-climate" viticulture areas [64,65]. Overall, the outcomes of this study integrate the debate on the impacts of climate change on viticulture in Europe, which so far, was mainly based on the paradigm that temperature will continue to gradually rise in the future.

**Supplementary Materials:** The following are available online at http://www.mdpi.com/2073-4395/9/7/397/s1, Figure S1: Relation between convection activity and temperature in the SPG, Figure S2: Anomalies (2069–2078 vs 2059–2068) of the occurrence of the phenological stages as simulated by CSIRO-Mk3-6-0 for Syrah according with the ensemble mean of the phenological models, Figure S3: Anomalies (2069–2078 vs 2059–2068) of the occurrence of the phenological stages as simulated by CSIRO-Mk3-6-0 for Cabernet sauvignon according with the ensemble mean of the phenological models, Figure S4: Anomalies (2069–2078 vs 2059–2068) of the occurrence of the phenological stages as simulated by CSIRO-Mk3-6-0 model for Grenache according with the ensemble mean of the phenological models, Figure S5: Anomalies (2069–2078 vs 2059–2068) of the occurrence of the phenological stages as simulated by CSIRO-Mk3-6-0 for Chardonnay according with the linear non-sequential model, Figure S6: Anomalies (2069–2078 vs 2059–2068) of the occurrence of the phenological stages as simulated by CSIRO-Mk3-6-0 model for Chardonnay according with the linear sequential phenological model, Figure S7: Anomalies (2069–2078 vs 2059–2068) of the occurrence of the phenological stages as simulated by CSIRO-Mk3-6-0 model for Chardonnay according with the curvilinear sequential phenological model, Figure S8: Evolution of the main phenological stages in four traditional winemaking regions based on the ensemble mean of climate projections, Figure S9: Evolution of the main phenological stages in six regions potentially involved in northward varietal shift based on the ensemble mean of climate projections, Figure S10: Anomaly (2069–2078 vs 2059–2068) in climatic suitability for premium production of Syrah, Figure S11: Anomaly (2069–2078 vs 2059–2068) in climatic suitability for premium production of Cabernet sauvignon, Figure S12: Anomaly (2069–2078 vs 2059–2068) in climatic suitability

for premium production of Grenache, Figure S13: Anomaly (2069–2068 vs 2010–2019) in climatic suitability for premium production of Chardonnay.

**Author Contributions:** Experimental design, G.S. and D.S.; methodology, G.S., D.S., I.G.d.C.-A., N.O. and C.v.L.; formal analysis, G.S. and D.S.; investigation, G.S.; writing-original draft preparation, G.S.; writing-review and editing, D.S., I.G.d.C.-A.; N.O. and C.v.L.

**Funding:** This research was funded by the French National Research Agency (ANR), grant number ANR-10-LABX-45.

**Acknowledgments:** We thank the French National Research Agency (ANR) for the financial support in the frame of the Investments for the future Programme within the Cluster of Excellence (Labex COTE). We are grateful to three anonymous reviewers for their valuable and constructive comments that improved the manuscript.

**Conflicts of Interest:** The authors declare no conflict of interest.

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
