# Peer review of "The Impact of Possible Decadal-Scale Cold Waves on Viticulture over Europe in a Context of Global Warming"

_agronomy, doi:10.3390/agronomy9070397_

Round 1
Reviewer 1 Report
This manuscript presents interesting predictions for future (possible) effects of climate changes which is including models which are usually not taken in consideration and could be very important for future development of viticulture in Europe.
Similar studies are too often not considering that despite the most probable trend of temperature increase in 21st century, we can also expect more and more probable challenging conditions and climate oscillations, which will probably have more intensive effects outside the currently suitable (traditional) vine growing areas of Europe. Decadal scale could waves induced by possible collapse SPG oceanic convection is one of the possible scenarios, and detailed insight on his effects on grapevine phenology presented in this study I find valuable for scientific audience, but also for wine industry in Europe.
Author Response
We are grateful to reviewer 1 for these nice comments, which evidence how the main message of our study has been clearly grabbed.
Reviewer 2 Report
The study demonstrates the need to consider measures and projection of climate change (global warming vs rapid cold waves) in planning for future investments in viticulture. The study shows the relevant impact of cold waves in such a decision-making process. It is an interesting and quite innovative approach. There are many limitations in the models but the author provides a clear explanation for each point. Language is ok and the paper is well organised.
The study deals more with the technical/scientific side of the problem but perhaps the economic or sustainability focus is missing. I suggest in the discussion to include a reflection on the economic impact or on the sustainability of the presented scenarios from a more broad policy perspective (stressing the policy need, what can be done.., what already exists.., how it could be done...)
Author Response
Thank you to reviewer 2. We appreciate this suggestion and added a wider discussion about the economic and sustainability impacts that possible cold waves over Europe may imply.
Form lines 510-526 “Taking into account, at the same time, the possibility of a long-term continuous warming and the possibility of rapid cold events poses further challenges in the planning of proper measures of adaptation for wine producing sector in Europe, and in the analysis of their economic impact and sustainability. Indeed, beyond the effective costs that the expected increase of temperature will presuppose for preserving the wine production over Europe, our results suggest a further economic evaluation of the risks associated with an amplified decadal variability. This implies the research of producing systems that are able to face, at the same time, the warming trend and the possibility of unprecedented large and rapid changes over Europe. Such an approach implies an accurate analysis of the sustainability of radical choices, thus promoting a more rational strategy of adaptation, e.g. a diversified variety relocation, which takes into account the risk of sudden temperature changes for each specific region. Also, this approach can partly resize the pursuit of new wine regions, notably in those regions appearing particularly sensitive to large inter-decadal variability. For example, according to our results, hypothetical new vineyard settlements would be more suitable in the south-eastern part of Europe rather than in its north-western part, since the latter would be primarily impacted by the occurrence of the cold waves originated in the North Atlantic. These considerations become notably relevant if one considers the large economic effort presupposing the potential settlement of new vineyards in cool climate regions.”
Reviewer 3 Report
General comments
The article describes the impact on viticulture of potential cold waves over Europe, projected by the CSIRO-Mk3-6-0 model for RCP4.5 scenario, that can be superimposed on long-term warming trend.
It is really interesting the connection between long-term warming and cold waves concerning phenological stages, fruit composition and then opportunity for new vineyards areas.
You analyzed some grapevines varieties, also currently cultivated in Italy.
Recently a set of long term simulations on a late ripening variety (cv. Nebbiolo), but performed in the past climate, over an area in Piedmont region in Italy, have been analyzed and described (Andreoli V et al., Description and Preliminary Simulations with the Italian Vineyard Integrated Numerical Model for Estimating Physiological Values,Agronomy 2019, 9, 94). Results about the warming trend are similar to those described in your paper also if the analysis have been carried out in the past.
Specific comments
Line 164: “Sim Simulation....” I think it is a mistake.
Line 261: the values of cardinal temperatures are suitable for all different varieties?
Table1: Why for Grenache cultivar are not present the values for topt and F*?
Line 269: Do
you assume the same temporal range for climatic suitability for all
varieties and areas? Why?
Figure2: In the figure, particularly for UK graph, two other cooling event are observable. Have you analyzed also these events?
Line 662 and 737: The name "van Leeuwen" is in bold
P { margin-bottom: 0.08in; }
